# Long Term Interactions of Native and Invasive Species in a Marine Protected Area Suggest Complex Cascading Effects Challenging Conservation Outcomes

**Charalampos Dimitriadis** [1,*], **Ivoni Fournari-Konstantinidou** [2], **Laurent Sourbès** [1], **Drosos Koutsoubas** [1,2] **and Stelios Katsanevakis** [2]

1   National Marine Park of Zakynthos, El. Venizelou 1, 29100 Zakynthos, Greece; info@nmp-zak.org (L.S.); drosos@aegean.gr (D.K.)
2   Department of Marine Sciences, Faculty of Environment, University of the Aegean, 81000 Mytilene, Greece; ivofk@yahoo.com (I.F.-K.); stelios@katsanevakis.com (S.K.)
*   Correspondence: xdimitriadis@marine.aegean.gr

**Abstract:** Understanding the interactions among invasive species, native species and marine protected areas (MPAs), and the long-term regime shifts in MPAs is receiving increased attention, since biological invasions can alter the structure and functioning of the protected ecosystems and challenge conservation efforts. Here we found evidence of marked modifications in the rocky reef associated biota in a Mediterranean MPA from 2009 to 2019 through visual census surveys, due to the presence of invasive species altering the structure of the ecosystem and triggering complex cascading effects on the long term. Low levels of the populations of native high-level predators were accompanied by the population increase and high performance of both native and invasive fish herbivores. Subsequently the overgrazing and habitat degradation resulted in cascading effects towards the diminishing of the native and invasive invertebrate grazers and omnivorous benthic species. Our study represents a good showcase of how invasive species can coexist or exclude native biota and at the same time regulate or out-compete other established invaders and native species.

**Keywords:** alien species; herbivores; niche theory; marine protected areas

## 1. Introduction

Biological invasions are largely recognized as a major threat to the marine realm worldwide that can induce the decline of native biodiversity and negative impacts on ecosystem functioning [1–3]. The most pronounced changes in the ecology of the invaded ecosystems include the loss of native genotypes, degradation of habitats, changes in trophic interactions, and displacement of native species [4]. Marine invasive species may have negative socio-economic impacts to coastal societies, affecting ecosystem services such as food provision, tourism and recreation [1]. Although the Mediterranean Sea is considered to be a hotspot of marine biological invasions, their effects are overlooked in marine conservation planning [5,6] and have not received proper attention in the European network of marine protected areas [7].

Marine ecosystems are impacted by a multitude of human stressors and climatic change effects acting in concert [8,9]. Networks of marine protected areas (MPAs) have emerged as an explicit management tool not only to address the ecological impacts of local stressors (e.g., fishing, marine traffic, sand extraction) but also to increase resilience to global stressors, such as climate change [10]. However, the interrelated effects between protection and biological invasions are not yet well understood, often resulting in contradicting results [11]. The high or low performance of invasive species has been mainly explained by contrasting mechanisms and hypotheses in invasion biology [12,13]. The "biotic resistance hypothesis" (i.e., ecosystems with high species richness are more resistant to invaders than

those with low biodiversity) [14] predicts that the high native species richness in MPAs could prevent the establishment of alien species or, if established, substantially reduce their ecological impacts. Additionally, the restoration of top-down regulation processes (i.e., restoration of top predators' populations) in MPAs could also contribute to the control of invasive populations [15]. Conversely, the "biotic acceptance hypothesis" predicts a positive interaction between the invaders and the native species [16,17] and therefore MPAs can enhance the spreading and the abundance of invasive species within their limits. Still, "the enemy release hypothesis" [18] predicts that invasion success relies upon the absence of enemies, hence harvesting closures in MPAs can also be beneficial for the populations of invasive alien species. On the other hand, 'neutral theory' highlights the importance of randomness in the structure of community assemblages, and concludes that both high- and low-diversity communities are equally susceptible to biological invasions [19].

Empirical evidence, albeit restricted, suggests both positive and negative effects of the protection on invasive species largely relying upon the species involved, the geographical region, the size of the MPAs and their protection level as well as human induced pressures, environmental conditions and other intrinsic features [20–22]. Trophic cascades, top-down control, competitive exclusion and niche portioning are among the mechanisms that can explain the native vs. invasive and invasive vs. invasive species interactions in the invaded ecosystems [23–25]. In any case, understanding and assessing the interrelation of MPAs and invasive species as well as species interactions is critical for the effective management of the MPAs particularly on the long run, since the long-term effects of biological invasions are rarely investigated and hard to predict [19,26].

Here we provide empirical evidence on community shifts in a Mediterranean MPA (National Marine Park of Zakynthos, Greece) impacted by biological invasions, by estimating occupancy (probability of presence) changes of both alien and native species belonging to different taxa (fish, macroalgae and invertebrates) and different trophic levels. To this end, we analyzed data, collected through occupancy surveys of alien and native species, in 2009 and 2019, in an effort to detect: (a) the long term changes in ecosystem structure of rocky reefs in the MPA, (b) the possible presence of trophic cascades, and (c) the interaction within and between invasive and native species. Finally we discuss our findings in the perspective of management of biological invasions in MPAs.

## 2. Materials and Methods

### 2.1. Study Area

This study was conducted at the National Marine Park of Zakynthos (NMPZ), which is located in the southernmost part of Zakynthos Island, eastern Ionian Sea, Greece (Figure 1), and hosts one of the most important rookeries of the threatened marine turtle *Caretta caretta* in the Mediterranean [27]. Acknowledging the prime ecological importance of this area, the NMPZ was formally established in 1999 (Presidential Decree 906D/1999) and encompasses 83.3 km$^2$ of marine protected area in Laganas Bay. The latter is partially divided into three seasonal sub-zones (A,B,C) that are characterized by varying levels of protection with respect to fisheries and other human activities affecting turtles (Figure 1). From May to October, Zone A is a no-take/no boating-access area were mild human activities (e.g., swimming, snorkeling, scuba diving) are allowed, while only small-scale artisanal fishing and boat traffic—without anchoring—is permitted in zone B, under a maximum 6 knot speed limit; during the same period, zone C shares the regulations of zone B, but anchoring is permitted. From November to April, the aforementioned restrictions are lifted, and all sub-zones are subjected to the general rules that apply year-round in the NMPZ, such as the prohibition of trawlers, purse seiners, recreational fishing and tanker vessels [28].

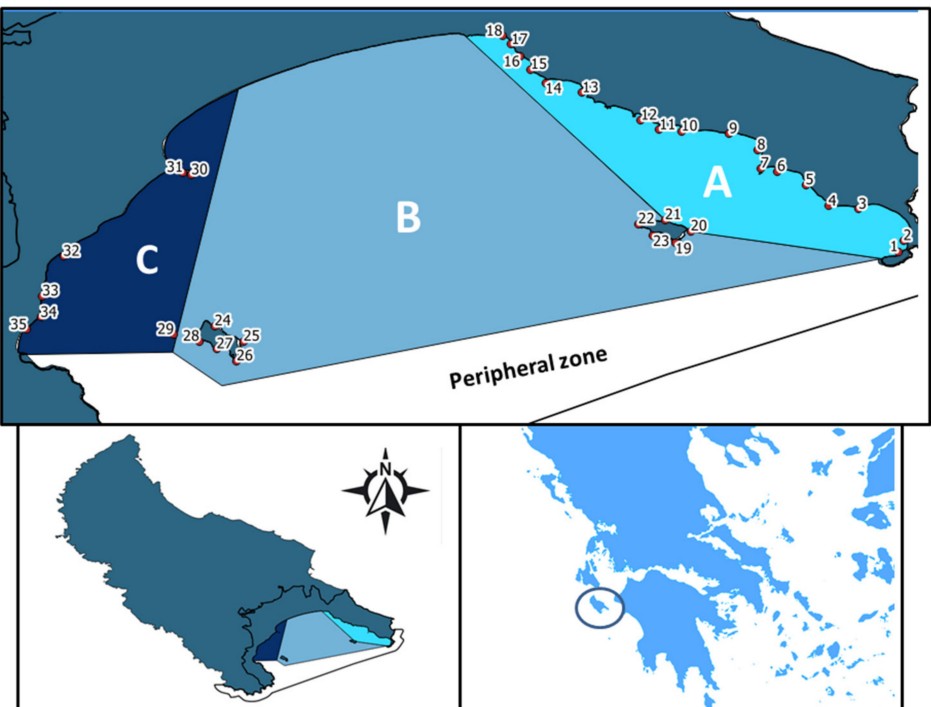

**Figure 1.** Study area and the 35 sampling sites at each zone (A,B,C and Peripheral) of the marine protected areas (MPA) of the National Marine Park of Zakynthos, Greece (NMPZ) that were sampled during 2009 and 2019.

### 2.2. Sampling Design

An important source of error in ecological monitoring is the inability of investigators to detect all individuals or even all species in surveyed areas, which is a common issue when monitoring populations in the marine environment [29,30]. If imperfect detectability is ignored, state variables (such as population density, abundance, probability of presence) are underestimated. Occupancy has been proposed as an appropriate state variable when monitoring the spatial and temporal evolution of biological invasions [29–31]. To address the issue of imperfect detection, the approach developed by MacKenzie et al. [32] of estimating occupancy, $\psi$, by jointly estimating detectability, $p$, based on repetitive surveys at each station, was followed, according to the protocols developed for marine surveys by Issaris et al. [30].

To estimate the occupancy of the target megabenthic and necto-benthic species along the shallow rocky reefs of NMPZ, 35 sampling sites were selected in a systematic way (with consecutive sites having a 200 m distance between them) along the rocky coastline of NMPZ. The distribution of rocky reefs among the three protection zones is driven by the geomorphology of the area (large sandy beaches between rocky coasts), and thus the spatial distribution of sampling sites was not uniform (Figure 1). The same sites had been surveyed previously by Thessalou-Legaki et al. [33], with exactly the same approach. At each sampling site, a 200 m transect was defined along the coastline by the use of a diving reel. Epibenthic megafauna was recorded (simple presence/absence) at each transect by two independent experienced observers, by snorkelling, during 20 min long surveys at each transect and at a depth range of 0–10 m under calm water conditions of high in-water visibility. Identification was conducted in situ and photographs were taken. The selected taxa that were considered in this study consisted of species of conservation importance, invasive species, or key species for rocky reefs (Table 1). Two species were surveyed only in 2019 (*Epinephelus costae* and *Paracentrotus lividus*) and thus occupancy was estimated only for that year. The same was done for the invasive fish *Siganus rivulatus*, which had

not yet invaded the MPA in 2009. All animals were assigned to trophic groups based on relevant scientific literature regarding species diet (Table 2).

**Table 1.** List of the targeted megabenthic and necto-benthic species (+ sign indicates whether a species is alien or native).

| | Alien | Native |
|---|:---:|:---:|
| **Arthropoda** | | |
| *Percnon gibbesi* (H. Milne Edwards, 1853) | + | |
| **Chordata** | | |
| *Siganus rivulatus* Forsskål & Niebuhr, 1775 | + | |
| *Siganus luridus* (Rüppell, 1829) | + | |
| *Epinephelus marginatus* (Lowe, 1834) | | + |
| *Epinephelus costae* (Steindachner, 1878) | | + |
| *Sparisoma cretense* (Linnaeus, 1758) | | + |
| **Rhodophyta** | | |
| *Lophocladia lallemandii* (Montagne) F.Schmitz, 1893 | + | |
| *Ganonema farinosum* (J.V.Lamouroux) K.C.Fan & Yung C.Wang, 1974 | + | |
| **Chlorophyta** | | |
| *Caulerpa cylindracea* Sonder, 1845 | + | |
| **Ochrophyta** | | |
| *Stypopodium schimperi* (Kützing) M.Verlaque & Boudouresque, 1991 | + | |
| **Echinodermata** | | |
| *Echinaster sepositus* (Retzius, 1783) | | + |
| *Ophidiaster ophidianus* (Lamarck, 1816) | | + |
| *Hacelia attenuata* Gray, 1840 | | + |
| *Paracentotus lividus* (Lamarck, 1816) | | + |

**Table 2.** Assignment of targeted species to trophic groups based on relevant scientific literature regarding species diet.

| Species | Trophic Group | Reference |
|:---:|:---:|:---:|
| *Percnon gibbesi* | benthic herbivorous invertebrate | [34] |
| *Siganus rivulatus* | herbivorous fish | [35] |
| *Siganus luridus* | herbivorous fish | [35] |
| *Epinephelus marginatus* | predatory fish | [36] |
| *Epinephelus costae* | predatory fish | [37] |
| *Sparisoma cretense* | herbivorous fish | [37] |
| *Echinaster sepositus* | benthic omnivorous invertebrate | [38] |
| *Paracentrotus lividus* | benthic herbivorous invertebrate | [39] |
| *Ophidiaster ophidianus* | benthic omnivorous invertebrate | [38] |
| *Hacelia attenuata* | benthic omnivorous invertebrate | [38] |

*2.3. Data Analysis*

The approach developed by MacKenzie et al. [32] is based on modeling the two stochastic processes that affect the outcome of whether a species is detected at a site. The species might either occupy a site (with probability $\psi$) or not (with probability $1 - \psi$). If the site is unoccupied, obviously the species will not be detected. If the site is occupied, at each survey $j$, the species will either be detected (with probability $p_j$) or pass undetected (with probability $q_j = 1 - p_j$). Hence, the probability $\mathbf{Pr}(\mathbf{H}_i)$ of any detection history $\mathbf{H}_i$ can be estimated as a function of $\psi$ and $p_j$, e.g., $\mathbf{Pr}(\mathbf{H}_i = 101) = \psi p_1 q_2 p_3$, where $\mathbf{H}_i = 101$ denotes that site $i$ was surveyed by three observers, with the species being detected by the first and third observers. Then, the likelihood of the data will be:

$$L(\psi, \, p|\mathbf{H}_1, \, \mathbf{H}_2, \dots, \mathbf{H}_s) = \prod_{i=1}^{s} \mathbf{Pr}(\mathbf{H}_i) \tag{1}$$

where *s* denotes the number of sites (=35). Covariates (such as the different year for $\psi$ or the observer for *p*) were included in the expression of the likelihood through the logistic model:

$$\theta_i = \exp(\mathbf{Y}_i\boldsymbol{\beta})\cdot(1 + \exp(\mathbf{Y}_i\boldsymbol{\beta}))^{-1} \tag{2}$$

where $\theta_i$ is the probability of interest (either occupancy or detection probability), $\mathbf{Y}_i$ are the covariates to be modeled, and $\boldsymbol{\beta}$ is the vector of the estimable covariate coefficients [32]. Standard maximum likelihood techniques were applied to estimate the model parameters.

The analysis of the data was conducted using the open-access software PRESENCE v2.13.6, using the multiple-season analysis engine [40], except for the species that were either surveyed only in 2019 or were absent in one of the two years, for which the single-season analysis engine was used. For occupancy we investigated whether it can be considered constant between the two time periods (2009 and 2019) or not. For detection probability, we investigated whether it differed by observer. A total of 4 models were fitted for each species. Model 1 assumed constant occupancy between the two years, and constant detection probability (independent of the observer—Null model). Model 2 assumed different occupancy between the two years, and constant detection probability. Model 3 assumed constant occupancy between the two years, and detection probability dependent on the observer. Model 4 assumed different occupancy between the two years, and detection probability dependent on the observer. For the species that were surveyed only in 2019 and those that were absent in one of the two years, only Models 1 and 3 were fitted using the single year with data availability. The estimation of occupancy was based on all plausible models following a multi-model inference approach on the basis of the small-sample, bias corrected form of the Akaike's information criterion ($\mathrm{AIC_c}$) [41]. The support of the hypothesis of differing occupancy between 2009 and 2019 (assumed in models 2 and 4) was assessed based on Akaike weights:

$$w_i = \exp(-0.5\Delta_i) / \sum_j \exp(-0.5\Delta_i), \tag{3}$$

where $\Delta_i = \mathrm{AIC_{c,}}_i - \mathrm{AIC_{c,min}}$ is the difference between the $\mathrm{AIC_c}$ value of model *i* and the one of the best model. Specifically, the support of this hypothesis was estimated by summing the Akaike weights of models 2 and 4 [41]. Model averaged estimates of occupancy were calculated by the formula:

$$\bar{\hat{\psi}} = \sum_i w_i \hat{\psi}_i \tag{4}$$

where $\hat{\psi}_i$ is the occupancy estimate by model *i* [41].

## 3. Results

Among the species which were present both in 2009 and 2019, in most cases model 2 (different occupancy, constant detectability) was the best (for *Siganus luridus*, *Echinaster sepositus*, *Ophidiaster ophidianus*, *Percnon gibbesi*, *Ganonema farinosum*), whereas model 4 (different occupancy, detectability varying by observer) ranked first in one case (*Epinephelus marginatus*), and model 3 (constant occupancy, detectability varying by observer) in another (*Sparisoma cretense*). The hypothesis of differing occupancy between 2009 and 2019 had > 90% support in the cases of *S. luridus*, *E. sepositus*, *O. ophidianus*, *P. gibbesi* and *G. farinosum* (Table 3).

**Table 3.** Best occupancy model for each target species, support of the assumption of differing occupancy between 2009 and 2019, and estimated occupancy (model-averaged) for the two years ($\psi$: occupancy; $p$: detection probability; obs: observer; n/a: not applicable).

| | Best Model | Support of the Assumption of Differing Occupancy between 2009 and 2019 | Occupancy | |
|---|---|---|---|---|
| | | | **2009** | **2019** |
| **Predatory Fish** | | | | |
| *Epinephelus marginatus* | $\psi$(period) $p$(obs) | 59.1% | 0.58 | 0.43 |
| *Epinephelus costae* [1] | $\psi$(.) $p$(.) | n.a. | n.a. | 0.09 |
| **Herbivorous Fish** | | | | |
| *Sparisoma cretense* | $\psi$(.) $p$(obs) | 28.7% | 0.69 | 0.73 |
| *Siganus rivulatus* *,[2] | $\psi$(.) $p$(.) | n.a. | 0.00 | 0.39 |
| *Siganus luridus* * | $\psi$(period) $p$(.) | 100.0% | 0.22 | 0.89 |
| **Benthic omnivorous invertebrates** | | | | |
| *Echinaster sepositus* | $\psi$(period) $p$(.) | 92.7% | 0.26 | 0.05 |
| *Hacelia attenuate* [3] | n.a. | n.a. | 0.06 | 0.00 |
| *Ophidiaster ophidianus* | $\psi$(period) $p$(.) | 92.4% | 0.64 | 0.16 |
| **Benthic herbivorous invertebrates** | | | | |
| *Percnon gibbesi* * | $\psi$(period) $p$(.) | 99.4% | 0.66 | 0.24 |
| *Paracentrotus lividus* [1] | $\psi$(.) $p$(.) | n.a. | n.a. | 0.09 |
| **Macroalgae** | | | | |
| *Caulerpa cylindracea* * | n.a. | n.a. | 1.00 | 0.00 |
| *Lophocladia lallemandii* * | n.a. | n.a. | 1.00 | 0.00 |
| *Stypopodium schimperi* *,[3] | n.a. | n.a. | 0.06 | 0.00 |
| *Ganonema farinosum* * | $\psi$(period) $p$(.) | 93.1% | 0.16 | 0.41 |

\* Alien species; [1] not surveyed in 2009; [2] absent in 2009 (it had not invaded the MPA yet); [3] modeling was not possible due to data scarcity (present only at two stations in 2009).

The model-averaged estimate for the occupancy of the native predatory fish *E. marginatus* indicated a 1.3-fold decrease from 2009 to 2019 (Figure 2, Table 3) (1.7-fold according to the best model); nevertheless, the possibility of constant occupancy cannot be excluded, as the hypothesis of differing occupancy had only 59.1% support, and actually all models had some support from the data. As regards the occupancy of the predatory fish *E. costae*, albeit not measured during 2009, particularly low occupancy levels were estimated in 2019. It has to be noted that for both species mostly juveniles were observed in 2019. The occupancy of all alien herbivore fish (*Siganus luridus* and *S. rivulatus*) substantially increased between 2009 and 2019, with *S. luridus* presenting a significant 4-fold increase, while *S. rivulatus*, from absent in 2009, it reached an occupancy of 0.39 in 2019. The benthic (native) omnivorous invertebrates *E. sepositus* and *O. ophidianus* displayed an abrupt occupancy decline between 2009–2019, since a 5.1- and 4.2-fold decrease was estimated, respectively. The omnivorous invertebrate *Hacelia attenuata* was present in two stations in 2009 but was not found in 2019. As regards the benthic herbivorous invertebrates, the invasive *Percnon gibbesi* exhibited an almost threefold decrease from 2009 to 2019. The native *Paracentrotus lividus* presented low occupancy values during 2019. The invasive macroalgae *Caulerpa cylindracea* and *Lophocladia lallemandii* were not found in any of the sampling sites during 2019, and thus their occupancy was considered as zero across the MPA. This is in marked contrast with the situation in 2009, when both species were present in all surveyed stations, and thus their occupancy was 1. *Stypopodium schimperi* was found at two stations in 2009 but it was completely absent in 2019. On the contrary the only macroalga that displayed a 2.5-fold increase in its occupancy from 2009 to 2019 was the alien *Ganonema farinosum*.

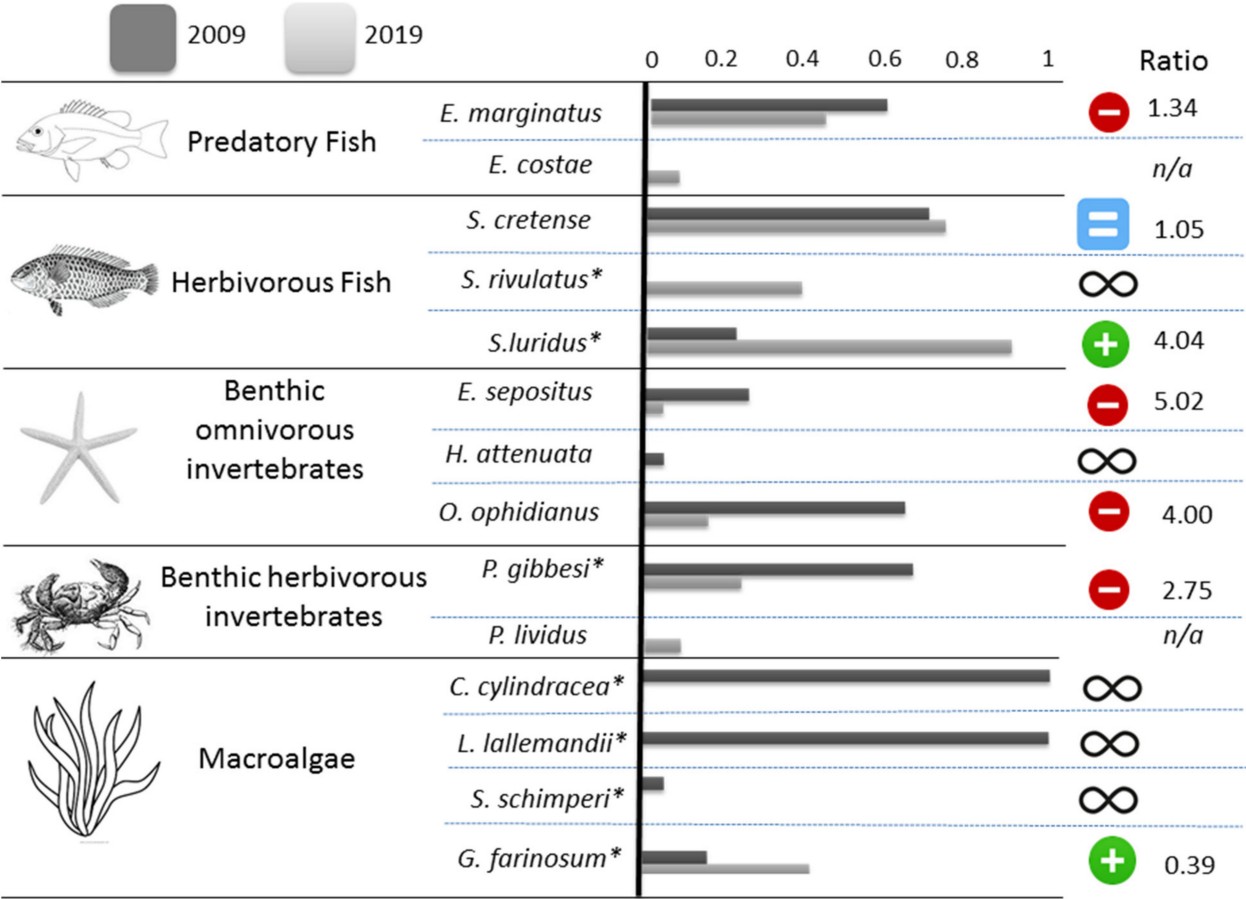

**Figure 2.** Species occupancy (model-averaged values) and the ratio between 2009 and 2019 occupancy values. n/a indicates species that were not examined during 2009 survey; asterisk (*) denotes alien species; the plus sign indicates an increase of occupancy and the minus sign a decrease (as indicated by the best model); the equals sign indicates no change while the infinity sign denotes infinite change due to a new invasion (i.e., *S. rivulatus*) or an extinction (*C. cylindracea, L. lallemandii, S. schimperi*).

## 4. Discussion

Here we found evidence of modifications in the rocky reef communities in the MPA, most probably due to the presence of invasive species altering the structure of the ecosystem and triggering complex cascading effects on the long term alongside with other intrinsic processes (Figure 3). The low populations of high-level predators could have contributed to the population increase and high performance of both native and invasive fish herbivores, thereby resulting in cascading effects towards the diminishing of the populations of the native and invasive invertebrate grazers and omnivorous benthic species. The latter was manifested mainly through resource limitation (i.e., benthic grazers) and habitat modification (i.e., benthic omnivorous species). The possible overgrazing induced by the dominant herbivores had pronounced negative effects on both native and invasive macroalgae, with the only exception being the invasive macroalgae *G. farinosum*, which is not consumed by the invasive and native herbivores. Our study represents a good showcase of how established invaders can either thrive [29,42,43] or fail [20,44] in an MPA through long term and complex species interactions. We also found evidence that invasive species can coexist with native biota and at the same time regulate or outcompete other established invaders and native species probably through top-down control and cascading effects.

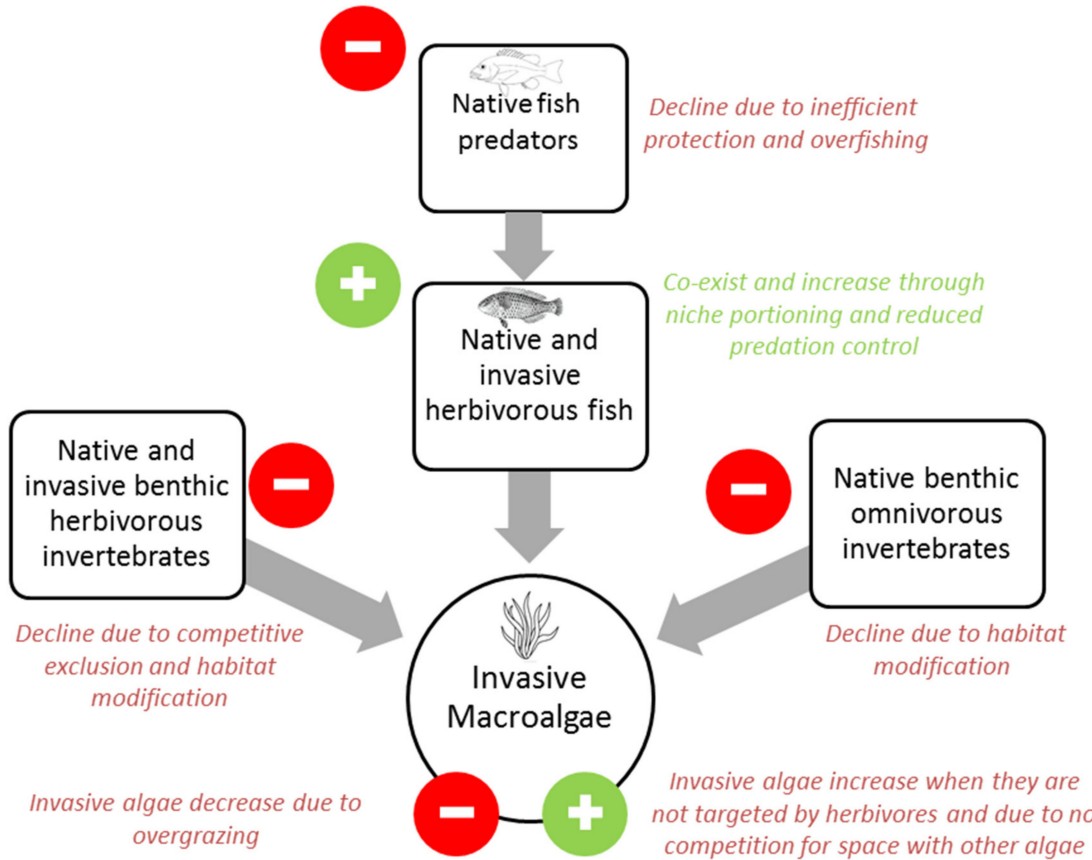

**Figure 3.** Schematic representation of the long term interactions between and within the native and invasive species in a Mediterranean MPA (− sign stands for a decline in the occupancy and + sing for an increase).

The possible decline of the occupancy of the dusky grouper *Epinephelus marginatus* as well as the low occupancy of *E. costae* during 2019, and the fact that only juveniles were observed, can be considered as indication of the insufficiency of the existing protection measures to adequately protect large carnivorous fish [28]. Recent studies in the MPA during 2012 and 2016 clearly demonstrated that apex predators are subjected to an overexploitation status, and their biomass and density were particularly low when compared to other Mediterranean MPAs [22,28]. The recurrent monitoring of the fishing pressure in the MPA by its personnel from 2010 to 2018 mainly suggests a downward trend in the fishing intensity (NMPZ unpublished data). The mass mortality outbreak of groupers during 2013 in the MPA due to viral nervous necrosis (VNN) (also known as viral encephalopathy and retinopathy (VER)) [45] may have contributed to the decline of the *Epinephelus marginatus* population considering also the slow growth rate and long life-span of this species [46].

Recent experiments have shown that high trophic level native predatory fish such as groupers can feed on alien herbivorous fish species such as Siganids and therefore the restoration of high-level predatory fish populations in Mediterranean MPAs can potentially exert top-down control on alien fish [47]. In the absence of substantial control, the substantial increase of the invasive herbivore fish *Siganus luridus* and *Siganus rivulatus* is of high concern. These species are finding a surprisingly favorable habitat in the Mediterranean Sea, in which shallow reefs are algae-dominated and occupied by only two potential herbivore fish competitors, *Sparisoma cretense* and *Sarpa salpa* [48]. The two Siganids are considered to be high-impact invasive species in the eastern Mediterranean Sea [1]. They have become dominant in many coastal areas [49,50] and alter the community structure and the native food web of the rocky infralittoral zone [49,51]. Based on a caging experiment, Sala et al. [49] concluded that *S. luridus* and *S. rivulatus* were able

to create and maintain barrens (rocky areas almost devoid of erect algae) and contribute to the transformation of the ecosystem from one dominated by lush and diverse brown algal forests to another dominated by bare rock. This rapid deforestation of algal forests at the rocky reefs in the eastern Mediterranean has been ascribed to the complementary roles of *Siganus luridus* and *Siganus rivulatus* with the first acting as a browser and the latter as a grazer [50] thus differentiating their niches and reducing competition for the limiting resource. In addition, *S. rivulatus* and *S. luridus* are very tolerant and adaptable species (i.e., temperature tolerance and diet plasticity), able to settle on a larger range of substrates and habitats than the native fish herbivores [35] and subsequently presenting higher fitness. The co-existence of the Siganids and *S. cretense* may arise as a result of low interaction strength and resource overlap between them, since *S. cretense* can graze more effectively on hard macrophytes [52] while the Siganids have a generalized diet including also invasive macroalgae [35,52]. The beneficial effect of protection upon the population of fish herbivores is in line with the results from previous studies in the MPA that reported considerably high abundance and biomass values of fish herbivores during 2012 [28] and 2016 [22] when compared to the total fish biomass in the MPA as well as to other MPAs in the Mediterranean [22] or elsewhere [53].

In 2001, it was reported that an important characteristic of the benthic flora at the MPA of the NMPZ was the dominance of *Cystoseira* species, forming very dense forests and supporting a rich associated flora on hard substrata [54]. *Cystoseira* spp. forests are ecologically very important as they support high biodiversity and act as nurseries for a number of demersal fish species [55,56]. Yet the invasive macroalgae *Lophocladia lallemandii* and *Caulerpa cylindracea*, albeit in low densities, were already established (first recorded during 2001) [54] in the MPA prior to the establishment of *Siganus luridus* and *Siganus rivulatus* which arrived around 2004 and 2014, respectively. In 2009, *Caulerpa cylindracea* was very abundant in NMPZ exhibiting an aggressive behavior on rocky reefs [57]. Since 2009 the gradual disappearance of *Lophocladia lallemandii* and *Caulerpa cylindracea* (which can be consumed by Siganids [35]) along with the gradual decline of macroalgae in the rocky reefs of the NMPZ has been observed, with the existence of barrens during 2019 being more frequent (unpublished data). Still a strong association between barrens and invasive herbivores has been recently evidenced for the MPA [22] or other areas in the eastern Mediterranean [50]. As regards *Ganonema farinosum* it was the only macroalgae that was observed to increase its occupancy. This species has been found to be positively affected by overgrazing by Siganids as it seems to be avoided by these herbivores, whereas competition for space and light with other macroalgae decreases [58,59].

The most prominent cascading effect of fish herbivore overgrazing (followed by the decline of top–down control by predators) was evidenced upon the populations of the benthic omnivorous sea stars (i.e., *Echinaster sepositus* and *Ophidiaster ophidianus*). These native sea stars experienced abrupt declines, most likely indirectly through habitat degradation (deforestation) and the consequent reduction of prey. The population dynamics of the sea stars has been previously correlated to algal coverage on Mediterranean rocky reefs [60]. Recent studies have demonstrated that *O. ophidianus* feeds mostly as a selective grazer that complements its diet by other animal organisms [61]. Likewise, the invasive benthic herbivore *P. gibbesi* has undergone a substantial decrease from 2009 to 2019 while the native benthic herbivore *P. lividus* presented very low occupancy levels during 2019 irrespectively of the protection status of the MPA. This finding is at odds with previously results that reported the presence of *P. lividus* at rocky reefs across the MPA at high densities during 2012 [62].

Despite the contradicting results regarding the role of marine protected areas upon the success of biological invasions (e.g., [11,63,64]), there is growing evidence that the established invaders could present higher fitness than the native species in the Mediterranean MPAs often resulting in complex cascading effects and altering the structure and function of the invaded ecosystems [11,22,43]. Yet our study moves one step forward by highlighting the importance of the interaction not only between native and invasive species

but also between invasive species. However, such generalizations should be still treated with caution since the available spatial and temporal information is limited.

There is growing consensus that invasion ecology is conceptually embedded within the unifying principles of ecology, and the mechanistic conceptual framework describing biological invasions should be assimilated therein [13]. Therefore, we seek out the mechanisms that can explain the long-term changes of the trophic web in MPAs due to biological invasions, through the lens of modern coexistence theory and contemporary niche theory (see [65] for a review). Coexistence or exclusion of native and invasive species largely depends upon the available niche space created by their interactions, while modern coexistence theory recognizes invasion winners and losers among the species of the recipient ecosystems [24]. In this respect when native vs. invasive and/or invasive vs. invasive species are competing for a limiting resource, either coexistence or competitive exclusion of the inferior competitor will occur, depending on niche similarity among competing species [66]. This means that the higher the fitness and niche differences between species, the more likely to coexist and produce indirect effects at varying competitive strength [25]. In other words, local community assembly relies on how species traits interact with community filters to formulate species abundance, and therefore alien species can either underperform or outperform native species [67].

As the impact of invasive alien species, favoured by climate change, seems to be severe, management actions (see [68]) for the control of invasive herbivores should be carefully considered as they challenge conservation objectives in MPAs. Among the most commonly proposed management actions against invasive species in the marine realm are their targeted removal and commercial and/or recreational utilization of specimens along with the restoration of their top-down control by native predators. Further research is still required in an effort to thoroughly understand the long term interrelation patterns between invasive species, native species and MPAs as well as to test the feasibility and efficacy of the proposed management measures against invasive species.

**Author Contributions:** C.D. and S.K. conceived and designed the study; C.D. and I.F.-K. conducted fieldwork; C.D., S.K. and I.F.-K. analyzed the data; C.D., S.K. and I.F.-K. wrote the manuscript; other authors provided editorial advice. All authors have read and agreed to the published version of the manuscript.

**Funding:** This study was funded by the project "Climate-Smart Coastal Practices for Blue governance-BLUECOAST", funded by the Interreg IPA II Cross-border Cooperation Programme "Greece–Albania 2014–2020" and National resources (Public Investment Programme—Greece).

**Institutional Review Board Statement:** Not Applicable.

**Informed Consent Statement:** Not Applicable.

**Data Availability Statement:** The data presented in this study is available on request from the corresponding author, because it is a part of an ongoing project.

**Conflicts of Interest:** The authors declare no conflict of interest.

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
