# Peer review of "Long Term Interactions of Native and Invasive Species in a Marine Protected Area Suggest Complex Cascading Effects Challenging Conservation Outcomes"

_diversity, doi:10.3390/d13020071_

Round 1

Reviewer 1 Report

This is an important and useful contribution not only to conservation purposes but also as a document that could (and should) be held by politicians and decision makers. Its importance well exceeds the island of Zakynthos and even the eastern Mediterranean basin since it does provide applicable methodologies and suggests improvements in the management of MPA's that certainly may well be applied elsewhere.

I congratulate the authors on this work.

Author Response

We would like to thank the reviewer for reviewing our manuscript

Reviewer 2 Report

The manuscript presented for review concerns the topic of long term interactions of native and invasive species in a marine protected area suggest complex cascading effects challenging conservation outcomes. It is an important and topical topic. Increasingly, the number of invasive species that interact with native species is increasing. This very often leads to the retreat of the native species, which is not a desirable process. The presence of complex cascading effects affects the maintenance results, which must be taken into account when creating protection plans and protective tasks.

In my opinion, the research was well planned, carried out and described by the Authors. The manuscript reads well and is communicative. The figures used match the text well, are legible and increase the quality of the manuscript.

The only thing missing for me is to relate the results to other marine ecosystems - can the observed relationships be regarded as "general truth" for marine protected areas? Could some algorithm be identified for marine protected areas that could be used by different researchers around the world? 

Author Response

Dear reviewer, we would like to thank you for your valuable comments. We have  now added a new paragraph in the discussion in an effort  to address your comment. The new text now reads: 

'Despite the contradicting results regarding the role of marine protected areas upon the success of biological invasions (e.g. [11, 63, 64]), there is growing evidence that the established invaders could present higher fitness than the native species in the Mediterranean MPAs often resulting in complex cascading effects and altering the structure and function of the invaded ecosystems [11,22,43]. Yet our study moves one step forward by highlighting the importance of the interaction not only between native and invasive species but also between invasive species. However, such generalizations should be still treated with caution since the available spatial and temporal information is limited'.